# Novel Robotic Arm Working-Area AI Protection System

**DOI:** 10.3390/s23052765

**Published:** 2023-03-02

**Authors:** Jeng-Dao Lee, En-Shuo Jheng, Chia-Chen Kuo, Hong-Ming Chen, Ying-Hsiu Hung

**Affiliations:** 1Department of Automation Engineering, National Formosa University, Yunlin County 632, Taiwan; 2Doctor’s Program of Smart Industry Technology Research and Design, National Formosa University, Yunlin County 632, Taiwan

**Keywords:** robotic arm, AI protection system, YOLO, object detection algorithms, working safety

## Abstract

From traditionally handmade items to the ability of people to use machines to process and even to human-robot collaboration, there are many risks. Traditional manual lathes and milling machines, sophisticated robotic arms, and computer numerical control (CNC) operations are quite dangerous. To ensure the safety of workers in automated factories, a novel and efficient warning-range algorithm is proposed to determine whether a person is in the warning range, introducing YOLOv4 tiny-object detection algorithms to improve the accuracy of determining objects. The results are displayed on a stack light and sent through an M-JPEG streaming server so that the detected image can be displayed through the browser. According to the experimental results of this system installed on a robotic arm workstation, it is proved that it can ensure recognition reaches 97%. When a person enters the dangerous range of the working robotic arm, the arm can be stopped within about 50 ms, which will effectively improve the safety of its use.

## 1. Introduction

With the rapid development in industry since 1950, old electrical appliances have been unable to meet operational needs, and things have gradually moved from electrification to automation, which is the so-called Industry 3.0. Technological advances include computers, aviation technology, robots, materials, etc. Epson corps. has even kept the production quality of each watch consistent since 1981 [1]. They began to develop robotic arms, and produced a series of industrial robotic arms for use in internal production lines. Industry began to use robotic arms a lot, and after years of accumulation, technology continued to improve. According to statistics from the International Federation of Robotics [2], as many as 1.7 million new robots were used in factories between 2017 and 2020. However, with the advancement of science and technology, industrial accidents also occur. An appalling industrial incident occurred in 1984 [3]. A middle-aged male operator with 15 years of experience in die-casting was operating an automated die-casting system. His body was trapped between the rear end of the industrial robot and the steel safety bar, resulting in his death. The hydraulic robot, which was used to remove die-cast parts from the die-casting machine and transfer those parts to the trimmer, had just undergone a one-week robotics training session three weeks before the fatal incident. According to the Bureau of Labor Statistics database of the fatal occupational injury census conducted by the National Institute for Occupational Safety and Health, from 1992 to 2015, there were 61 fatal incidents related to robots [4], and not only in the United States. According to the occupational accident statistics of the Labor Insurance Bureau of Taiwan in 2020 [5], occupational accidents in the previous five years had killed 300 people. The workplace fatalities in the manufacturing industry are second only to the high-risk construction industry. In an analysis of manufacturing disasters, as shown in Figure 1, 21.97% involved a worker becoming trapped in machinery and 16.22% comprised serious injuries from other causes. While machinery brings convenience, it also comes with a lot of risks. It can be seen from Figure 2 that the most commonly injured bodily injury sites are the finger, hand, and foot. If there is a lack of industrial safety knowledge for emerging equipment, accidents will continue in the workplace.

## 2. Related Works

Industry 4.0 and artificial intelligence have completely changed the traditional manufacturing model. In order to improve the production of goods on a large scale, or at least diversify procedures, digital processing and the introduction of AI are the development goals of various industries today. With the increase in collaborative work between robotic arms and human beings, hidden industrial safety risks, such as being struck, cut, and trapped follow. Therefore, the main focus of this research was to develop a robotic arm working-area AI protection system to provide a safer working environment. The system will involve AI visual algorithms: YOLO, user interface design, traditional electronic fences, and industrial safety.

### 2.1. Convolutional Neural Networks, CNN

When conducting AI visual detection research, a convolutional neural network (CNN) is one of the best choices for research. Its structure is clear. It is composed of one or more convolutional layers and connected layers, and it performs quite well in image detection. There are different opinions on the development history of CNN. It should be traced back to 1960, when Hubel and Wiesel [6] came to many conclusions when they studied the visual cortex of cats. They inserted microelectrodes into the visual cortex of an anesthetized cat, making it immobile. Line images of different angles were placed in front of the cat. Through the microelectrode information, it was found that some neurons would fire rapidly when viewing lines at a specific angle, while other neurons responded best to lines at different angles and classified the visual cortex. For both simple and complex reasons, some of these neurons respond differently to light and dark patterns, which led to the concept of CNN.

LeCun [7] proposed the LeNet-5 model to imitate the neural structure of the visual cortex, in which the convolution layer, subsampling layer, and fully connected layer are the basic elements for building the current CNN model. The structure has six layers after deducting the input and output (the convolutional layer of C1), the pooling layer of S2, the convolutional layer of C3, the S4 pooling layer, the C5 convolution and the F6 fully connected layer. This model is used to recognize handwriting and font, known as the ancestor of CNN. So far, it has been used in various types of image recognition problems and inspired the extension and development of various types of CNN models, such as ImageNet, AlexNet the and visual geometry group (VGG). These models have achieved high recognition rates in recognizing images.

In the industry of technology, Imoto, Nakai et al. [8] studied a defect analysis task that requires engineers to determine the cause of yield degradation from defect classification results during the inspection process of semiconductor manufacturing. The analysis is divided into three stages: defect classification, defect trend monitoring and detailed classification. To support the analysis work of the first- and third-stage engineers, a CNN using the transfer learning method was used for automatic defect classification. The proposed method is evaluated on a real semiconductor manufacturing dataset by performing the defect classification task using SEM and thoroughly examining its performance, which can classify defect images with high accuracy while reducing labor costs equal to that required for manual inspection of one third of the labor.

In agriculture, Chhaya Narvekar and Madhuri Rao [9] solved the dilemma of detecting flowers due to the wide variety of flowering plants, which are similar in shape, color and appearance. The prototype CNN model architecture and transfer learning method were proposed, and the flower classification of Kaggle’s flower dataset was checked on the architectures of VGG16, MobileNet2 and Resnet50. The flower classification is widely used and can be applied to field monitoring, plant identification, medicinal plant, flower industry, and plant taxonomy research.

### 2.2. YOLOv4

After understanding the basic architecture of CNN, we can discuss related extension algorithms. J. Redmon [10] and others published the first version of “YOLO” in 2015. Like its full name, object detection is regarded as a single regression problem, and only one-stage CNN is used to complete object detection, which has a very good recognition speed.

Since the development of YOLO, many versions have also improved it, including YOLOv2, YOLOv3, YOLOv3-tiny, YOLOv4, YOLOv4-tiny, YOLOv5, etc. Among them, YOLOv4, established by Alexey et al. in 2020 [11], is used by many people. The architecture of the backbone is improved by YOLOv3. After inputting photos, the original backbone: Darknet53 is imported into CSPNet. This method is usually used in low-end computers. After adding it, it can make the model lightweight, reduce memory consumption and maintain accuracy. In the neck part, SPP and PANet are used. The former significantly increases the receptive fields, isolates key features and maintains the network without reducing the running speed. PANet is improved based on the feature pyramid networks (FPN) used in YOLOv3, adding an extra layer with the original parts. Head follows the single-layer detection of YOLOv3. Compared to training EfficientDet object detector or NAS-FPN object detector, only a single graphics card with GPU can train the model, which saves a lot in cost.

### 2.3. YOLO-Related Application Literature

In YOLO-related applications, Dewi et al. [12] compared generative adversarial networks (GANs) such as DCGAN, LSGAN and WGAN to create more diverse training images, combining synthetic images with original images to enhance the dataset and validate the effectiveness of the synthetic dataset and comparing the original image model with the synthetic image model. They achieved 84.9% accuracy on YOLOv3 and 89.33% accuracy on YOLOv4. In terms of agriculture, Mamdouh et al. [13] proposed an improved deep learning framework based on YOLO to detect and count the number of olive fruit flies, which used data augmentation to increase the dataset size, include negative samples in training to improve detection accuracy, and normalize images to a yellow background to unify lighting conditions. The results of the proposed framework show a mean average precision (mAP) of 96.68%, which is significantly better than existing pest detection systems. In the coal mining industry, Xu et al. [14] proposed a cloud collaboration framework for real-time intelligent video surveillance in the coal mine underground environment. The framework realizes model optimization, and can still realize data between edge devices and the cloud for scenarios with poor network environment. transmission, and proposed the FL-YOLO instant object detection model based on YOLO. Compared with traditional video surveillance methods, it has excellent immediacy and accuracy. Qin et al. (2022) [15] showed excellent detection of major objects that drivers are concerned about during driving. It does not detect all of the objects appearing in the driving scenes, but only detects the most relevant ones for driving safety. It can largely reduce the interference of irrelevant scene information, showing potential practical applications in intelligent or assisted driving systems. Zhao et al. (2022) [16] proffered a video-based fire detection system with improved “you only look once” version 4 (YOLO-v4) and visual background extractor (ViBe) algorithms. By using the simplified Bi-FPN with attention mechanism in place of the PANet feature fusion network in the YOLO-v4 algorithm, the system can perform better at fire detection. In the past, AI-related research rarely focused on industrial safety topics, so this paper will innovatively apply the YOLO model to identify personnel and related equipment and use it in factories for personnel safety protection.

### 2.4. Industrial Safety-Related Literature

Nowadays, in order to increase the safety of operators when operating machines, large manufacturers need to develop more products in this regard, including Keyence [17], Omron [18] and Banner [19], etc. The most commonly used safety devices are safety light curtains and safety-door switches. The working principles of the two are slightly different depending on the brand. Safety light curtains consist of several light receivers and light emitters. The light receivers receive infrared rays from the light emitters. If the light receivers cannot receive signals within the receiving range, it will determine which group of light receivers are blocked and send a signal to remind the user. Safety-door switches are mostly mechanical or electromagnetic switches with the common goal of protecting people and machines from injury.

The hardware structure of safety light curtains is quite intuitive and usually consists of a group of cylinders. In addition to the basic light receiver and light emitter, a function similar to that of a rectangular grating can be achieved by adding a reflector. There are also special functions for users to check the height, prominence, quantity and other special functions of items, which are widely used. The shape of the safety-door switch is similar to a door lock. It must be installed on the outer doorframe of the machine during installation, which is more limited in use, but also because of its hardware nature, it is safer than the noncontact type of safety grating. This is mostly used for high-speed, large or dangerous machines with inertial rotation.

## 3. Materials and Methods

### 3.1. Industrial Safety-Related Literature

YOLOv4-tiny is the successor of YOLOv4 developed by Chien-Yao Wang et al. [20]. It is designed for low-end GPU devices. The neural network architecture of the backbone uses CSPDarknet53-tiny instead of YOLOv4 CSPDarknet53, which uses the CSPBlock module instead of the ResBlock module in the residual network, and can enhance the CNN learning ability. Neck chose YOLOv3-tiny’s FPN and LeakyReLU activation instead of the Mish activation function in YOLOv4. After comparing the performance of YOLOv3, YOLOv4, YOLOv3-tiny, YOLOv4-tiny in mAP and frames per second (FPS) by Jiang et al. [21], the performance is shown in Table 1, where YOLOv3 and YOLOv4 perform better than the tiny version in terms of mAP, so they are more suitable for cloud or fixed devices, while the tiny version has fast FPS performance and is suitable for embedded systems. Among them, the mAP performance of YOLOv4-tiny is nearly 8% better than YOLOv3-tiny.

Based on the detection speed, it was decided to use YOLOv4-tiny as the object detection algorithm in this study, and the next step is to decide which environment to use to train the YOLOv4-tiny model. At present, mainstream training sets such as Microsoft’s COCO and the annual PASCAL VOC challenge have rich human training sets, but this study expects a large number of robotic arms and machine photos to be verified in the factory. Therefore, in this study, by making models according to different fields and collecting field photos in a customized way, a large degree of accuracy can be achieved with a small number of photos. In this study, the grinding station of the laboratory factory is used as an example. The camera is installed above to collect photos. A lot of unnecessary noise can be avoided in the photo collection, and the actual recording of the actual situation has met the above customization. The standard is for large photos but large amounts of accuracy. After selection, we did not choose traditional PyTorch or TensorFlow, but used Darknet developed by YOLO original Joseph Redmon. Darknet is an open source neural network framework written in C and compute unified device architecture (CUDA). Although it is not popular, its structure is clear and Python can be used for training. Characteristics such as the weights file from it make many people choose Darknet as the training network. The training process is shown in Figure 3. The webcam image is put into the computer and divided into a large number of photos: 80% of the photos are used for training, and 20% are used for testing. The test is used to improve the accuracy, and then LabelImg is used to frame the objects of all photos and store the default XML file. Since the format read by YOLO is .txt, it must convert the training and test data from XML to .txt, the ratios of which match the photo ratio. Finally, it is put into Darknet for training. After training to get the weight file, the user can put it into the program for detection.

### 3.2. Personnel Location Identification Algorithm (PLIA)

The personnel location identification algorithm (PLIA) can determine whether a person is in a dangerous zone or not. If we want to determine whether a person has entered the warning range, there is no way for only YOLO to do so. As such, we combine YOLOv4-tiny with PLIA, detect the position of the object through YOLOv4-tiny, and then transmit the position of the object to PLIA for determining whether the worker is safe. The object detected by YOLOv4-tiny can be marked by a rectangle, and four vertex coordinates will be obtained. We define it from left to right and from top to bottom as subscripted a–d in Figure 4. The four vertices of the dangerous area are *Ra*, *Rb*, *Rc*, and *Rd*. The four vertices of the warning area are *Ya*, *Yb*, *Yc*, and *Yd*. The four vertices of the detected object are *Da, Db*, *Dc*, and *Dd*.

A basic judgment of whether the person is in a safe area is to determine the four points *Da, Db, Dc,* and *Dd* overlapping the *Ra*, *Rb*, *Rc*, *Rd* or *Ya*, *Yb*, *Yc*, *Yd*. If only overlapping comparisons are made between individual two regions, cross- and continuous output will be generated when a person enters these two areas at the same time.

The flowchart of the algorithm is shown in Figure 5. The dangerous range and warning range will be set as red and yellow, respectively. This will determine whether the detected object is a person, and then start to determine the location of the person and when the person crosses the two areas. It will determine that the person is in a more dangerous area, and will not determine that the person exists in two areas at the same time. Because the boxes are all rectangles, it only takes two judgments via the algorithm to determine whether the person is in the dangerous range. If not, it continues to determine whether they are in the warning range. Special attention should be paid to the relationship between the two ranges. The dangerous range is within the warning range, and it is necessary to avoid the warning range by means of subtraction when determining the warning range so that the coordinates will not be repeatedly determined. The corresponding output can be divided into three cases shown in Figure 6 is described as follows:In case A, since the *Da* of the person is in the dangerous range, the algorithm will determine them as being in the dangerous warning range, and will not enter the other determinants.In the case of part B, it first determines whether the person’s *Da, Db, Dc, Dd* are in the dangerous range. If not, it will determine whether they are in the secondary alert range. In B, the person’s *Da* are all within the warning range. It is determined to be in the warning range.In the case of part C, it first determines whether the person’s *Da, Db, Dc, Dd* are in the dangerous range. If not, it determines whether they are in the warning range, and finally whether they are currently in the safe range.

### 3.3. AI Protection Module Application

This system needs to be installed on a computer capable of computing AI, and most of today’s NVIDIA display cards are used as the core of computing AI. The GPU of NVIDIA display cards’ performance can possess superior image processing capabilities than traditional CPU in image processing capabilities. However, the general computer communication interface is mostly Ethernet or USB. On some traditional machines, the communication interface may already have a corresponding device, and then it will be inconvenient to use.

An NVIDIA-embedded system, Jetson AGX Xavier was adopted to solve the connection problem. More I/O can be connected than ordinary computers. It also has UART pins that can be used. The practicality of nontraditional and modern machines has been greatly improved and the computing performance will vary depending on the Jetson version, but basic image processing has considerable capabilities.

The actual hardware configuration is shown in Figure 7. The computing module uses Jetson AGX Xavier as the core. It can use various communication functions to implement many different combinations during control, including Socket, I/O, UART, Modbus, etc. To display the AI-determined result to the user, this research purposely connects the controlled party to the programmable logic controller (PLC), which is quite common in the factory. As an extension to machine control, it is connected to the robotic arm controlled by this study and connected to stack lights to display the dangerous range and the warning range.

## 4. Experimental Results

### 4.1. System Actions and Architecture

In this study, the developed robotic arm working-area AI protection system will be applied to the automated KUKA robotic arm grinding station. The AI computing module is connected to the PLC via network communication and takes relays as a bridge between AI computing module and the robotic arm. As can be seen in Figure 8, the Jetson AGX Xavier with a camera is set up above the KUKA robotic arm, and the relay is derived through the network type PLC to control the robotic arm movement and display the processing status with the stack light.

The operation process of the robotic arm working-area AI protection system is divided into “UI program” and “AI program stages,” shown in Figure 9. Open the UI screen in the Jetson AGX Xavier with camera. The user account and password must be entered in this screen. If that is wrong, the terminal will display “Sorry, please try again.” Otherwise, it will enter the set UI screen if the account and password are correct. After entering, the screen contains many functions: the communication part can set the serial port, baud rate, byte size, parity bit, and stop bit. After the connect button is pressed, the system will detect whether the serial port number used is correct. If it is wrong, it will display “Error port, please try again” on the terminal. If it is correct, it can send or receive signals. To set the warning range, first preview the real-time image through the camera. The user can set the warning range according to the proportional scale marked around the screen and input the dangerous range (red) and the warning range (yellow) into the UI shown in Figure 10. After the input is completed, click the “Red OK” or “Yellow OK” buttons to display the two ranges on the screen. If the input range is not what was originally expected, user can enter the new range in the UI and click “Red OK” or “Yellow OK” button to view the range. In addition, the camera screen can be opened by clicking the “Open” button in the User interface to confirm the status of the actual robotic arm workspace shown in Figure 11. If confirmed, click “Confirm” to run the robotic arm working-area AI protection system.

In order to prevent danger, the robot arm will not be able to move when the program is not opened, and the initialization output must be added at the front of the process before releasing the emergency stop of the robot arm and the red state of the stack light, as shown in Figure 9. After the above state is released, the program will read the range value that needs to be warned. The reading of the warning value must first bring the input ratio value back to the actual pixel value according to the ratio and open the camera in Gstreamer format and create pipelines and component formats through the Gstreamer framework to increase the possibility of subsequent development in image processing. When the camera is opened, the M-JPEG server is opened immediately, and this server enables developers to upload images to it. Users can view the image by entering the server IP in any browser. The system will repeatedly detect whether the person is in the warning range, as shown in Figure 9. In this process, in addition to detecting the position of the robot arm and the person, it will also detect whether the person has entered the warning range. When detecting, the system will give priority to determining whether the person is within the dangerous range (red). If not, it will detect whether person is within the warning range (yellow). If the above conditions are met, the green light will be turned on if the person is not in both ranges.

### 4.2. Actual Action—Open Robotic Arm Workstation

In this section, the KUKA robotic arm grinding station is used as the implementation station. The user can use the “ifconfig” command to query the IP of the NVIDIA Jetson AGX Xavier, and enter the IP and port number in the browser. After the connection is successful, the system will start to detect and determine. When the autonomous mobile robot (AMR) enters the warning range or the dangerous range, the system will detect the AMR and turn on the green light to indicate that the AMR can grasp the object, as shown in Figure 12 and Figure 13. When the person enters the warning range, the system will light up the yellow light to warn the person, as in Figure 14. If the person enters the dangerous range, the robot arm will be stopped urgently and the red light will be turned on, as shown in Figure 15, and the result will be recorded in the logfile folder for that day. The logfile contains text files and screenshots, such as in Figure 16, allowing users to check the current situation after the event to verify whether there has been a misjudgment or an accident, and if it is safe, the green light will turn on, as shown in Figure 17. When the light source is insufficient or the lens is dirty, the image quality will be reduced, leading to lower recognition rates. Just like general visual inspection systems, it is necessary to ensure the quality of camera images to avoid misjudgments caused by improper image acquisition.

### 4.3. Comparison with Commercial Keyence Safety Grating

This research has successfully improved the problems of low hardware installation flexibility of general safety gratings, inability to quickly change the warning range, and the inability of autonomous mobile robots to enter the working area of the robotic arm. However, the developed robotic arm working-area AI protection system is actually different from general safety gratings. The image computing speed in the system is shown in Figure 18. it takes about 2–6 ms to detect from one image to the next, and the overall detection response time is about 23–51 ms. The response time of the safety gratings sold by Keyence is shown in Table 2. It will vary slightly with the length of the output cable. The response time of the safety light grid is still better than that of the work-area warning system developed in this study.

## 5. Conclusions

An AI protection system for a robotic arm in a specific working area was successfully developed in this study. This system can quickly adapt to different working areas and equipment. According to the experimental results, when a person enters the designated dangerous area, the robotic arm will stop within 51 s, and the abnormal time and image will be recorded for subsequent judgment. Moreover, users can set the warning and dangerous areas by themselves, which solves the problem of low installation flexibility of fixed electronic fences, the inability to quickly change the range, and the inability of AMR to work with robotic arms. The identification AI model is built through YOLOv4-tiny, which can be quickly built and run on edge devices, and the identification accuracy rate in this system is as high as 97%. Combined with the self-developed PLIA, when personnel enter the warning area, the red warning light will be on and the robotic arm will stop. This system solves the problems of traditional fixed electronic fences, including the large size and the inability to freely install and modify on the production line, the inability to communicate with old machines, the inability to connect with various arms, and the inability of AMR.

While the AI model used in this study was YOLOv4-tiny, actual use will not be limited to a specific AI model. As long as it is an AI model that can identify objects, mark their coordinates, and then utilize the PLIA proposed in this study to transmit the operation results to the robotic arm controller, it can carry out safety protection during robotic arm work. This experiment uses a network-type PLC to transmit operation results (work/stop signals) to the robotic arm controller, mainly considering the stability of PLC in industrial applications. Besides, remote I/O, microcontroller, or other communication transmission modules can also be used.

## Figures and Tables

**Figure 1 sensors-23-02765-f001:**
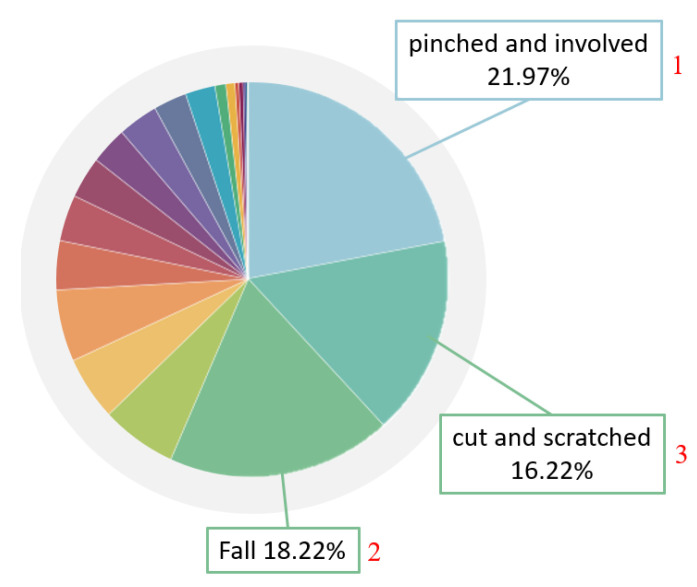
Occupational accident statistics of the Labor Insurance Bureau of Taiwan in 2020 [5].

**Figure 2 sensors-23-02765-f002:**
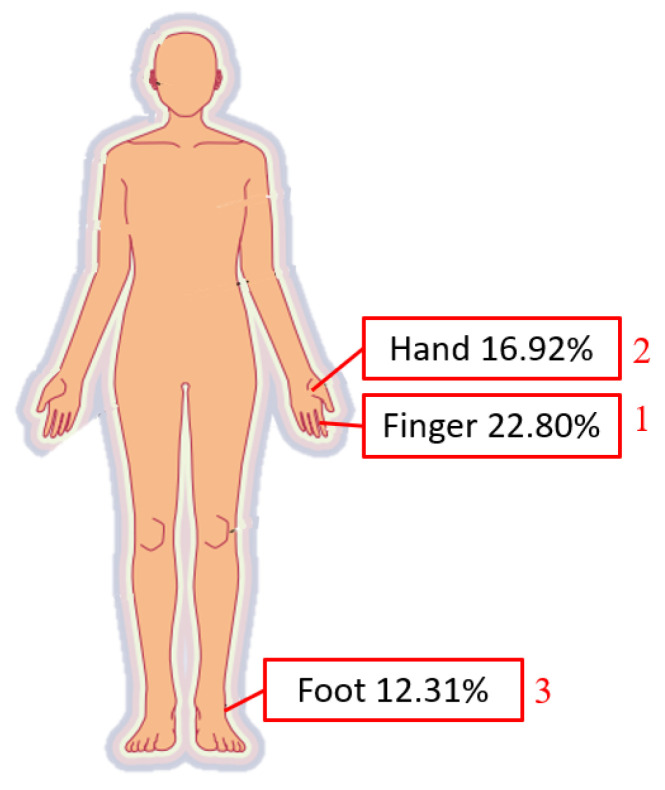
Bodily injury sites, from occupational accident statistics of the Labor Insurance Bureau of Taiwan in 2020 [5].

**Figure 3 sensors-23-02765-f003:**
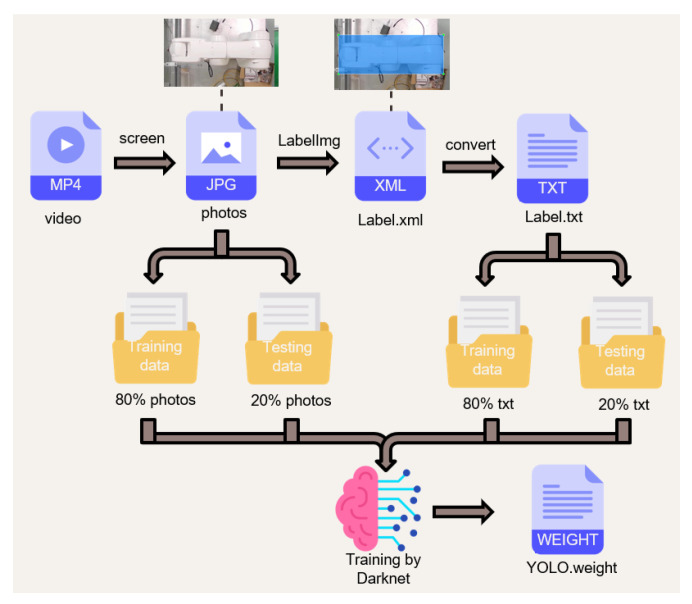
Customized dataset training steps.

**Figure 4 sensors-23-02765-f004:**
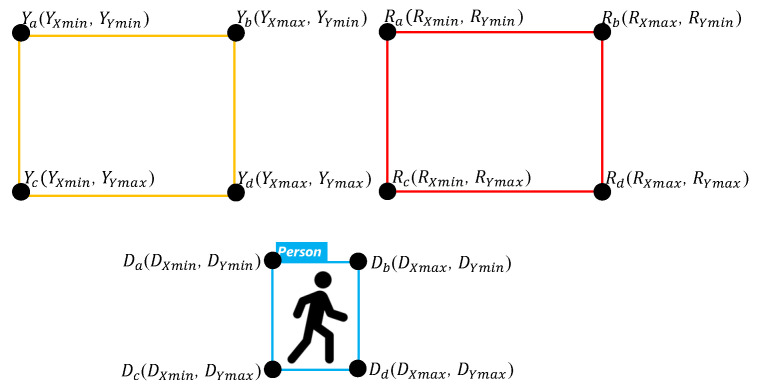
Rectangle detected by YOLO.

**Figure 5 sensors-23-02765-f005:**
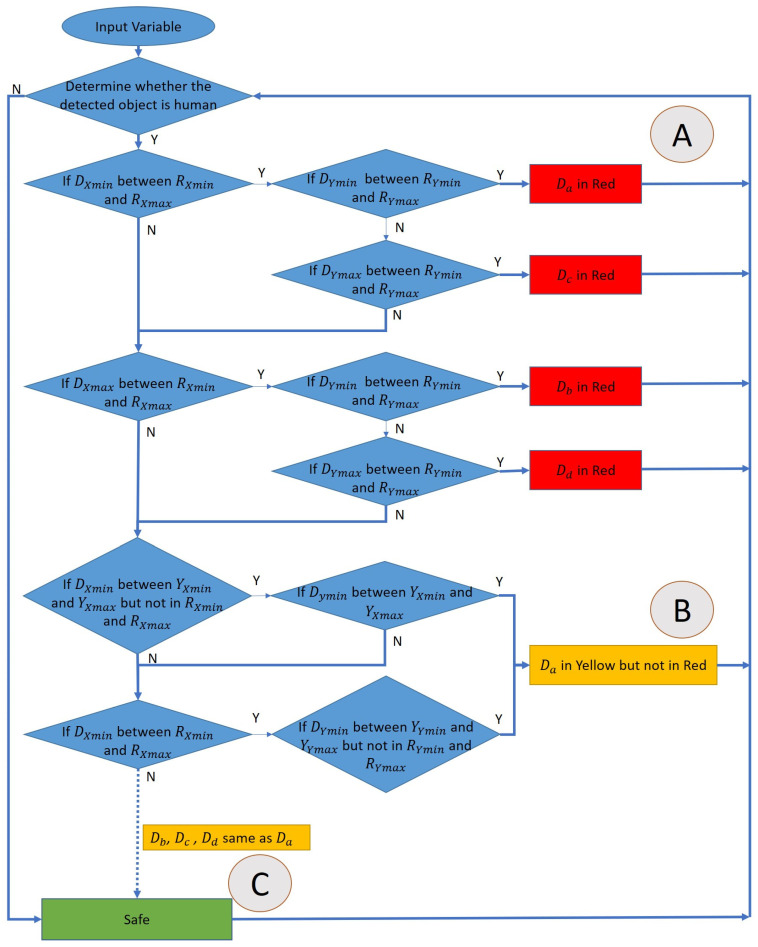
The flowchart of the algorithm.

**Figure 6 sensors-23-02765-f006:**
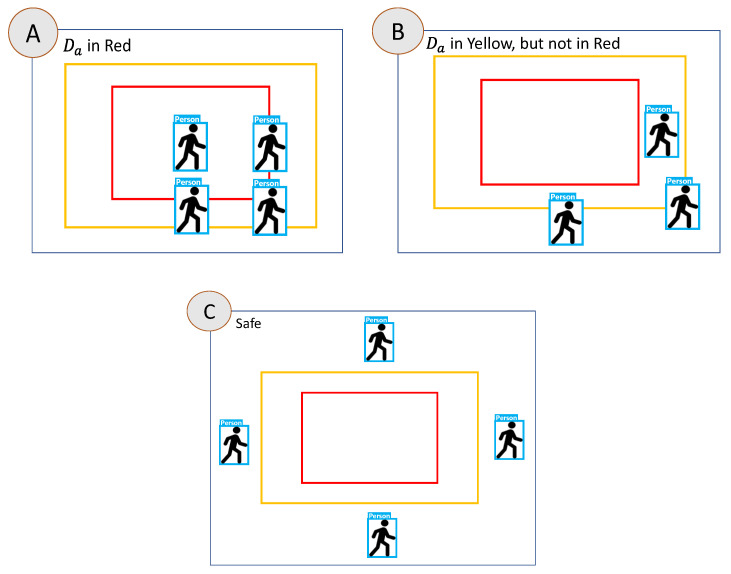
Case A, Case B and Case C.

**Figure 7 sensors-23-02765-f007:**
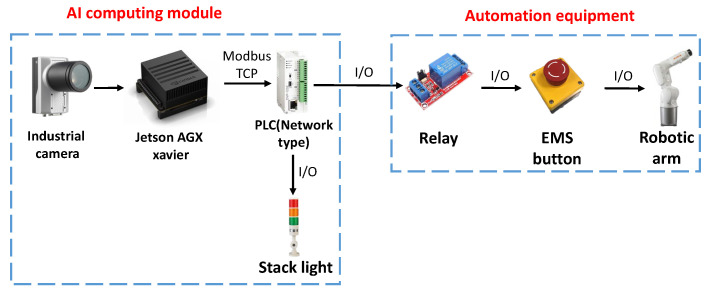
The actual hardware configuration.

**Figure 8 sensors-23-02765-f008:**
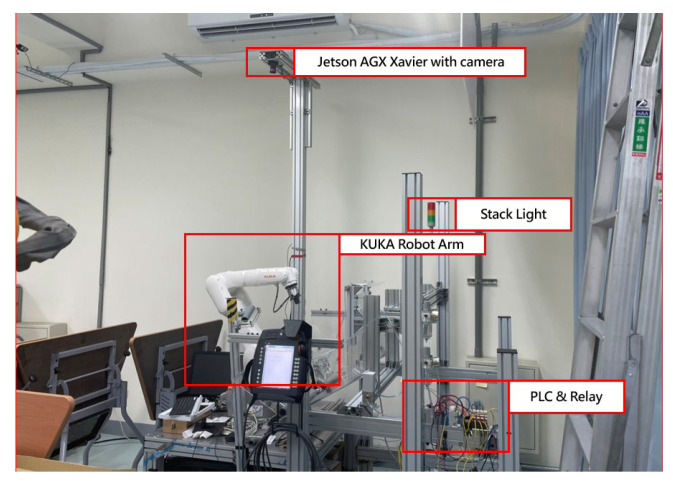
Install the AI module on the workstation.

**Figure 9 sensors-23-02765-f009:**
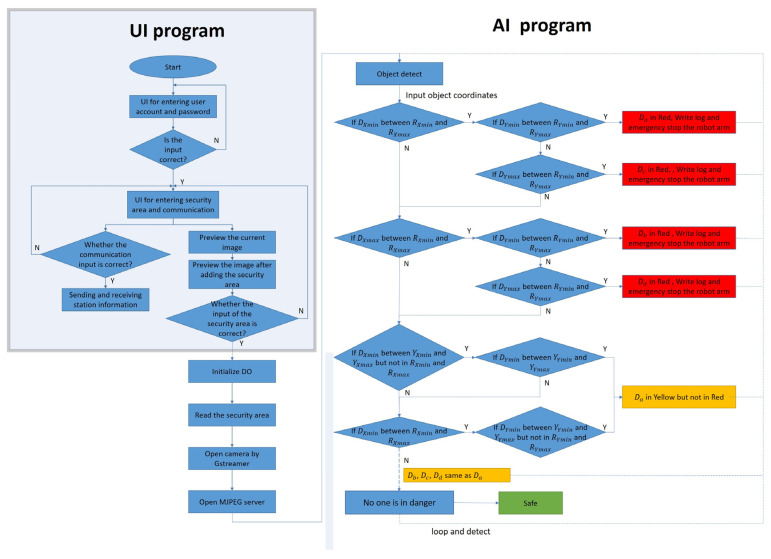
The operation process of robotic arm working-area warning system.

**Figure 10 sensors-23-02765-f010:**
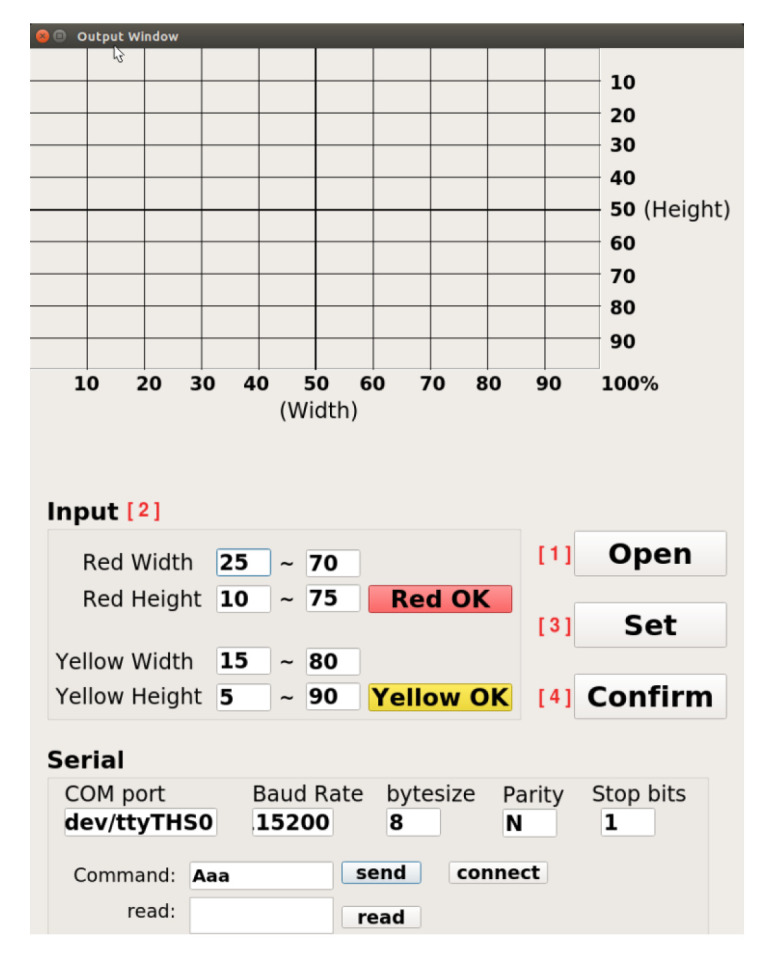
User interface for communication and input warning range functions.

**Figure 11 sensors-23-02765-f011:**
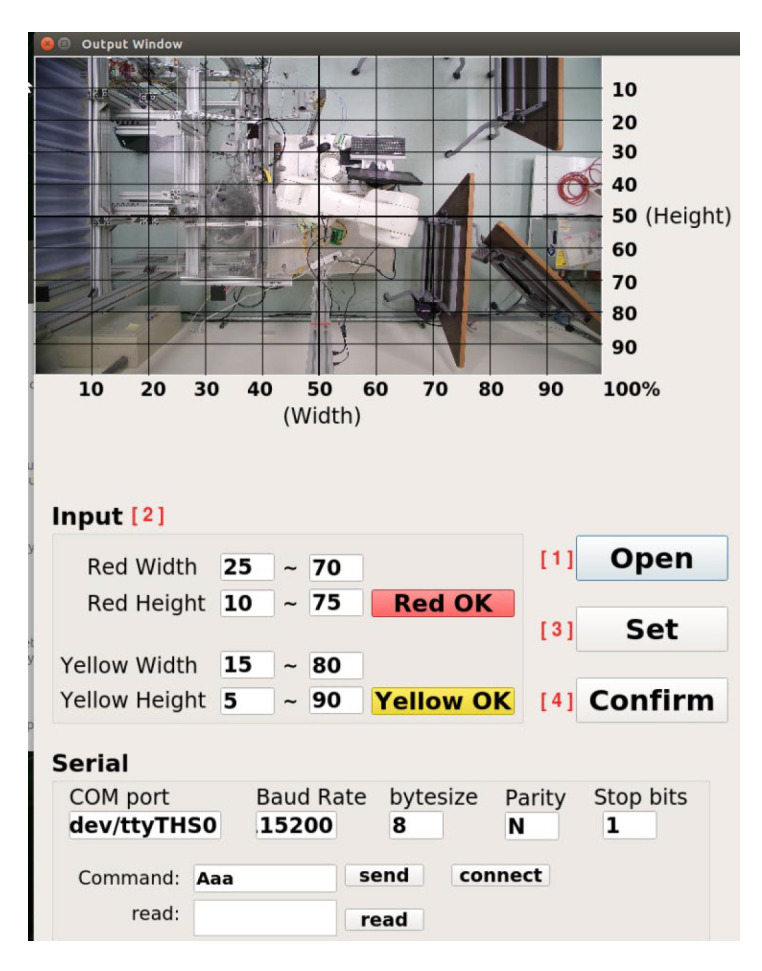
User interface after clicking the “open” button.

**Figure 12 sensors-23-02765-f012:**
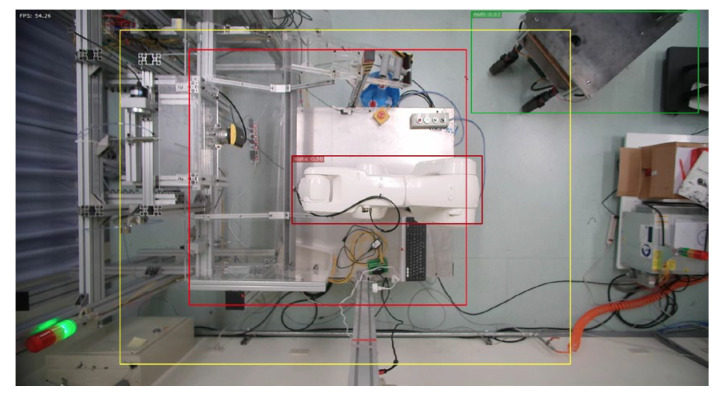
AMR enters warning range.

**Figure 13 sensors-23-02765-f013:**
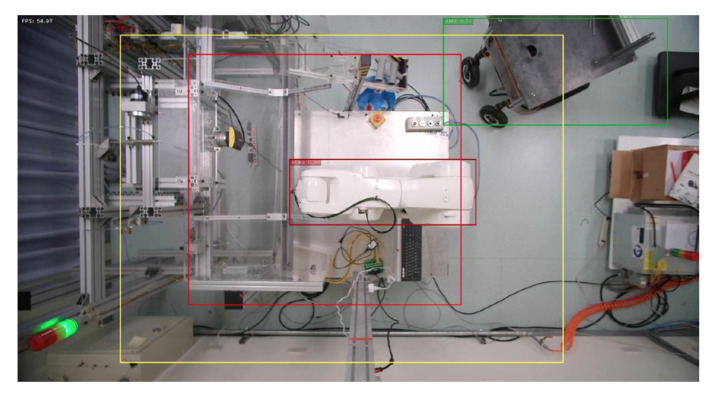
AMR enters first warning range.

**Figure 14 sensors-23-02765-f014:**
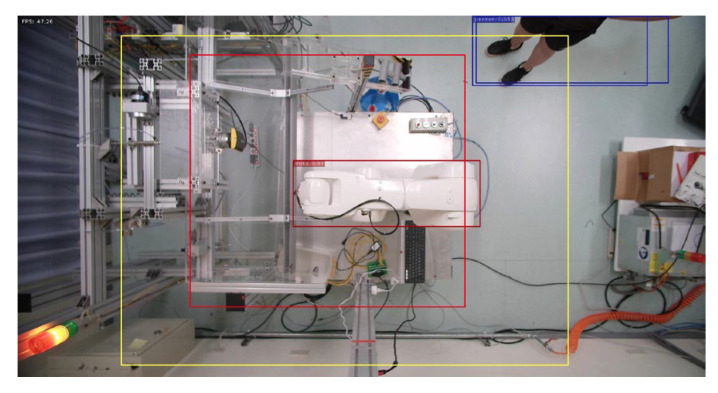
Person enters warning range.

**Figure 15 sensors-23-02765-f015:**
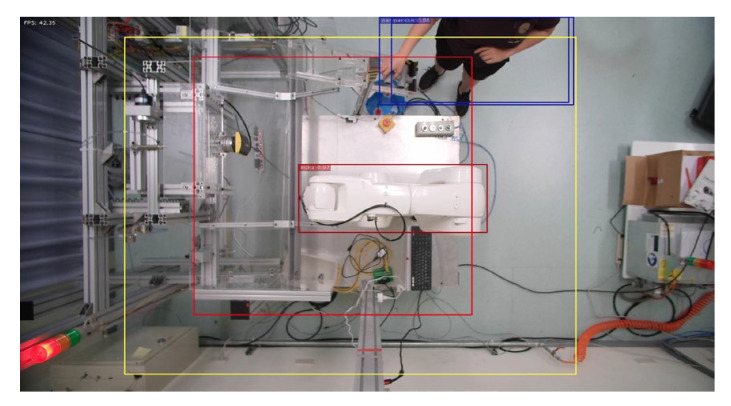
Person enters dangerous range.

**Figure 16 sensors-23-02765-f016:**
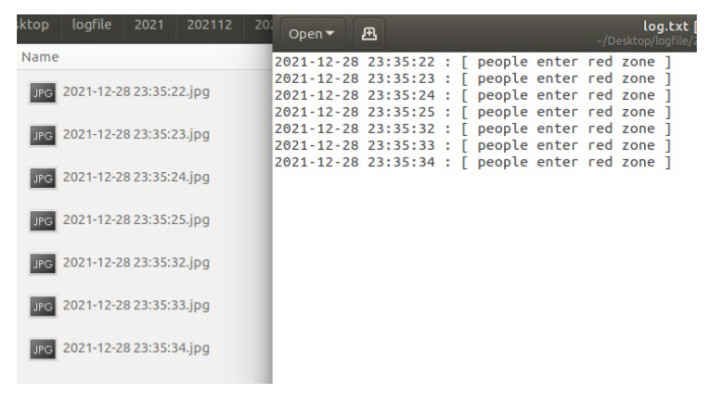
Contents of the logfile.

**Figure 17 sensors-23-02765-f017:**
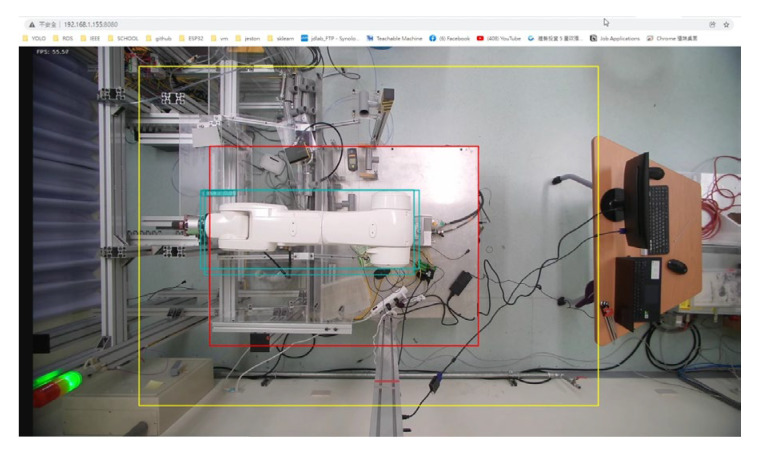
System shows safe when no object detected.

**Figure 18 sensors-23-02765-f018:**
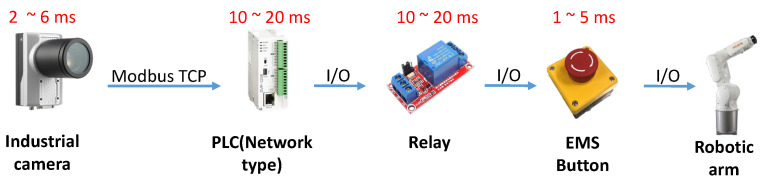
Overall detection time.

**Table 1 sensors-23-02765-t001:** YOLO comparison chart [21].

Method	FPS	mAP (%)
YOLOv3	49	52.5
YOLOv4	41	64.9
YOLOv3-tiny	277	30.5
YOLOv4-tiny	270	38.1

**Table 2 sensors-23-02765-t002:** Keyence safety light grid response time.

Type		Response Time (ms) On to Off
Mesh Type	Mesh Type	
GL-S08SH	GL-S08FH	6.6
GL-S12SH	GL-S12FH	6.6
GL-S24SH	GL-S24FH	7.0
GL-S36SH	GL-S36FH	8.3

## Data Availability

The author at the Mechatronics and Automation Laboratory, located at the Department of Automation Engineering, National Formosa University, Taiwan, was the subject of the experiments. The author consented to participate in this research study.

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
