# Peer review of "Novel Robotic Arm Working-Area AI Protection System"

_sensors, 2023, doi:10.3390/s23052765_

Round 1

Reviewer 1 Report

I have gone through the manuscript “Development and Application of a Robotic Arm Working Area AI Protection System”. I found it interesting and useful, but there are many flaws in the manuscript.  These need to be addressed.

My recommendation is to accept the article with following modification.

1.      Title of article needs to be clear according to work.

2.      Abstract is generic and need to add with specific information 

4.      There are various models available, how the current study is different from others.

5.      Many figures are blurred and need high resolution images.

6.      Figure 5 Flow chart need to be updated and should end with ellipse. Same as figure 9.

7.      All word should be added first instead of abbreviation.  

8.      Authors have presented their results and there is no critical discussion. Authors need to add results with detail discussion before the conclusion.

9.      Bench mark table should be added by giving the compression of current study with literature by giving its pros and cons.

1.  References are inadequate and need to add more recent literature. 

Author Response

Thank you for your valuable comments. The summary of changes is in the attached file. 

Reviewer 2 Report

Please consider the following comments:

1) the introduction is too long. almost 6 pages.

2) figure 5 is blurry, please use a high-quality image. The text is not aligned and centered and is inconsistent.

3) What are some of the limitations of this technique?

4) the system is dependent on the quality of the input image/video. Did the authors test the framework in a low-light environment?

5) what is the accuracy of the framework and what can affect the system the most?

Author Response

(The authors gave the same response as above.)
